# ScanSAR Interferometry of the Gaofen-3 Satellite with Unsynchronized Repeat-Pass Images

**DOI:** 10.3390/s19214689

**Published:** 2019-10-28

**Authors:** Zaoyu Sun, Anxi Yu, Zhen Dong, Hui Luo

**Affiliations:** College of Electronic Science and Technology, National University of Defense Technology, No. 109 Deya Road, Changsha 410073, China; yu_anxi@nudt.edu.cn (A.Y.); dongzhen@nudt.edu.cn (Z.D.); luohui@nudt.edu.cn (H.L.)

**Keywords:** Gaofen-3 satellite, ScanSAR, interferometry, interferometric coherence, phase compensation, DEM geolocation

## Abstract

Gaofen-3 is a Chinese remote sensing satellite with multiple working modes, among which the scanning synthetic aperture radar (ScanSAR) mode is used for wide-swath imaging. synthetic aperture radar (SAR) interferometry in the ScanSAR mode provides the most rapid way to obtain a global digital elevation model (DEM), which can also be realized by Gaofen-3. Gaofen-3 ScanSAR interferometry works in the repeat-pass mode, and image pair non-synchronizations can influence its performance. Non-synchronizations can include differences of burst central times, satellite velocities, and burst durations. Therefore, it is necessary to analyze their influences and improve the interferometric coherence. Meanwhile, interferometric phase compensation and rapid DEM geolocation also need to be considered in interferometric processing. In this paper, interferometric coherence was analyzed in detail, followed by an iterative filtering method, which helped to improve the interferometric performance. Further, a phase compensation method for Gaofen-3 was proposed to compensate for the phase error caused by the unsynchronized azimuth time offset of image pair, and a closed-form solution of DEM geolocation with ground control point (GCP) information was derived. Application of our methods to a pair of Gaofen-3 interferometric images showed that these methods were able to process the images with good accuracy and efficiency. Notably, these analysis and processing methods can also be applied to other SAR satellites in the ScanSAR mode to obtain DEMs with high quality.

## 1. Introduction

Launched on 10 August 2016, Gaofen-3 is a Chinese high-resolution remote-sensing satellite with a C-band multi-polarization synthetic aperture radar (SAR) payload [1]. Since then, it has been widely used in ocean surveillance, land management, ship detection, disaster reduction, and so on [2,3,4,5,6,7,8]. It can also be used with the SAR interferometry technique to extract a digital elevation model (DEM) of the Earth. SAR interferometry utilizes image phases, which contain topographic information, to obtain three-dimensional coordinates of the Earth’s surface. Because of its outstanding performance, it has become an important DEM mapping technique.

Gaofen-3 works in a sun-synchronous orbit, and its altitude is about 755 km. The revisiting period of Gaofen-3 is 29 days. Gaofen-3 can work in many working modes with different resolutions and swath characteristics, such as stripmap mode, spotlight mode, and scanning synthetic aperture radar (ScanSAR) mode. In the spotlight mode, the resolution is 1 m and the swath is 10 km × 100 km. In the ultra-fine stripmap mode, the resolution is 3 m and the swath is 30 km. In the standard stripmap mode, the resolution is 25 m and the swath is 130 km. In the narrow ScanSAR mode, the resolution is 50 m and the swath is 300 km. In the wide ScanSAR mode, the resolution is 100 m and the swath is 500 km. Among these modes, the ScanSAR mode is important as it can achieve wide-swath SAR images. SAR interferometry in ScanSAR mode can be used for wide-area topographic mapping because of this capability. This technique is worthy of in-depth research as a rapid global DEM-mapping method. In SAR interferometry, at least two images are needed, and this paper only considered two. The two SAR images used for Gaofen-3 interferometry are achieved in a repeat-pass mode.

For spaceborne remote sensing toward the Earth, the ScanSAR mode was first used in Spaceborne Imaging Radar-C (SIR-C) to acquire several experimental data. The SIR-C system was installed on a space shuttle and the mission was carried out in 1994 [9]. The Canadian satellite RADARSAT launched in 1995 was the first spaceborne SAR system with an operational ScanSAR mode [9]. Subsequently, SAR interferometry in ScanSAR mode has been deeply studied and widely used. The concept of ScanSAR interferometry was proposed in 1995 by Guarnieri [10]. He detailed ScanSAR interferometry and verified the interferometric method using simulated ERS-1 SAR data [11]. Bamler presented a ScanSAR interferogram using real RADARSAT data for the first time in 1999 [12], and in 2002, a complete description of RADARSAT ScanSAR interferometry was published [13]. In 2000, the Shuttle Radar Topography mission (SRTM) was carried out to map the world’s landmass. This project demonstrated the rapid mapping ability of ScanSAR interferometry, which was able to map the landmass of the Earth in 10 days [14]. SAR interferometry in ScanSAR mode has also been used in other satellites, such as ENVISAT [15,16], ALOS [17], ALOS-2 [18], and TerraSAR-X [19]. The Gaofen-3 satellite can also work in ScanSAR mode, and it is necessary to study its interferometry. In the above studies, the master and slave images used the same observing parameters. However, in Gaofen-3 ScanSAR interferometry, the images are unsynchronized and may have different pulse repetition frequencies (PRFs), velocities, and burst durations. These differences, together with the burst central time difference, influence the interferometric coherence. It is necessary to analyze these influences and present a corresponding filtering method to improve the interferometric coherence. Between the master and slave images, the unsynchronized azimuth time offset causes a phase error when there is no phase adjustment during imaging; thus, interferometric phase compensation is needed. This compensation is a problem that has not yet been studied. From the compensated interferometric phase, we can determine the DEM. In DEM geolocation integrated with the absolute phase calculation and calibration, the most efficient method is to determine a closed-form solution. It is necessary to derive a closed-form solution for DEM geolocation combined with absolute phase calculation and phase error compensation.

This paper discusses several questions in Gaofen-3 ScanSAR interferometry, and is divided into seven sections. Section 2 analyzes the interferometric performance of Gaofen-3 in ScanSAR mode. Section 3 presents the iterative filtering method to improve interferometric performance. Section 4 proposes a compensation for the interferometric phase in Gaofen-3 ScanSAR interferometry. Section 5 derives a closed-form solution of DEM geolocation with ground control point (GCP) information. Processing experiments with Gaofen-3 interferometric images in ScanSAR mode were made to verify the analyses and methods in Section 6. Conclusions are drawn in Section 7.

## 2. Interferometric Model and Performance of Gaofen-3

The ScanSAR mode is a SAR mode with a width swath. By beam scanning, ScanSAR can observe several sub-swathes simultaneously. These sub-swathes are located at different points along the range direction. Together, they can cover a whole wide swath. Because only a single beam is used in ScanSAR, the observing time must be separated and allocated to different sub-swathes. Thus, for a single sub-swath, the observing signals are in the burst mode. During bursts, signal pulses are transmitted to the sub-swath, but no signal pulses are used between bursts. In burst mode, the azimuth resolution is decreased. ScanSAR can cover a width swath but with low resolution. Thus, ScanSAR is suitable for rapid mapping, but not suitable for subtle measurement. ScanSAR interferometry is based on the ScanSAR mode, so it has similar characteristics.

The ScanSAR interferometry of Gaofen-3 works in a repeat-pass mode. The two images achieved from the two passes may be unsynchronized. In this paper, we used a pair of Gaofen-3 interferometric images over Kunlun Mountain. These images were achieved in ScanSAR mode for wide-swath remote sensing. The main parameters are listed as Table 1.

In the parameters, the PRFs and velocities are different. Burst durations are decided by PRFs and pulse numbers, so they were also different. Because there was no burst synchronization between the two images, a burst central time difference also existed. These unsynchronized characteristics influence interferometric performance.

From the ScanSAR principle, the ScanSAR mode observes the Earth’s surface only in bursts. It is different from the normal stripmap mode, which uses continuous observation. Thus, the interferometric performance of the ScanSAR mode needs to consider these burst characteristics in the signal model. The burst characteristics also include the above-mentioned unsynchronizations. This paper analyzed the interferometric performance of the ScanSAR signal model [20]:(1)$$\begin{array}{l} s(t,\tau )=\sum_{n=0}^{N}rect\left(\frac{t-T_{c}-nT_{d}}{T_{b}}\right)\cdot \iint_{D}V_{d}(r_{p},t_{p})W(t-t_{p}-t_{x})\exp\left[-j\frac{4\pi }{\lambda }R(r_{p},t-t_{p})\right]\\ \;\;\cdot a\left[\tau -\frac{2R(r_{p},t-t_{p})}{c}\right]\cdot \exp\left\{j\pi k{\left[\tau -\frac{2R(r_{p},t-t_{p})}{c}\right]}^{2}\right\}dr_{p}dt_{p} \end{array}$$
where
(2)$$R(r_{p},t-t_{p})=\sqrt{r_{p}^{2}+{\left[v_{r}(t-t_{p})\right]}^{2}}$$
where t is the slow time, τ is the fast time, $V_{d}$ is the scattering coefficient, W(·) stands for azimuth envelope, a(·) stands for pulse envelope, rect(·) stands for rectangular function, $r_{p}$ is the vertical distance from the orbit to target p, $t_{p}$ is the moment when the vertical sight line passing target p, $t_{x}$ is the time offset caused by squint, $v_{r}$ is the equivalent velocity, $T_{c}$ is the burst central time, $T_{d}$ is the burst cycle time, $T_{b}$ is the burst duration, and *k* is the pulse modulation rate.

The ScanSAR signal can be processed by the extended chirp scaling (ECS) algorithm [20,21,22]. In this algorithm, the signal is first translated into the range-Doppler domain and processed along the range dimension, and then processed by azimuth scaling and focusing. The range processing includes chirp scaling, bulk range cell migration correction (RCMC), range compressing, and second-range compressing. After range processing, we retrieve the processed signal in the range-Doppler domain. Considering a single burst, the processed signal can be expressed as follows [20,21,22]:(3)$$\begin{array}{ll} S(f_{t},\tau )= & \iint_{D}V_{d}(r_{p},t_{p})A\exp(-j2\pi f_{t}t_{p})\mathrm{sinc}\left[kT_{r}\left(\tau -\frac{2r_{p}}{cD(f_{t_{ref}})}\right)\right]\\ & \cdot W_{a}(f_{t}-f_{t_{x}})W_{b}(f_{t}-f_{t_{c}})\exp\left(-j\frac{4\pi r_{p}f_{0}D(f_{t})}{c}\right)dr_{p}dt_{p} \end{array}$$
where
(4)sinc(x)=sin(πx)/(πx)
(5)$$D(f_{t})=\sqrt{1-\frac{c^{2}f_{t}^{2}}{4v_{r}^{2}f_{0}^{2}}},\;W_{a}(f_{t})=W\left(\frac{-cr_{p}f_{t}}{2f_{0}v_{r}^{2}D(f_{t})}\right),\;W_{b}(f_{t})=rect\left(\frac{-cr_{p}f_{t}}{2f_{0}v_{r}^{2}D(f_{t})T_{b}}\right)$$
(6)$$f_{t_{x}}=-\frac{2v_{r}^{2}f_{0}t_{x}}{c\sqrt{r_{p}^{2}+v_{r}^{2}t_{x}^{2}}},\;f_{t_{c}}=-\frac{2v_{r}^{2}f_{0}(T_{c}-t_{p})}{c\sqrt{r_{p}^{2}+v_{r}^{2}{\left(T_{c}-t_{p}\right)}^{2}}},\;f_{t_{ref}}=f_{t_{x}}$$
$\mathrm{where}\;f_{t}$ is the azimuth frequency, A is a constant coefficient, $T_{r}$ is the pulse duration of the transmitted signal, and $f_{0}$ is the central frequency of the chirp signal.

By azimuth scaling processing, the second phase term of the range processed signal can be transformed as follows:(7)$$\exp\left(-j\frac{4\pi r_{p}f_{0}D(f_{t})}{c}\right)\underset{\to }{\;azimuth\;scaling\;}\exp\left(-j\frac{4\pi r_{p}f_{0}}{c}\right)\exp\left(j\frac{\pi r_{p_{ref}}c}{2v_{r}^{2}f_{0}}f_{t}^{2}\right)$$
where $r_{p_{ref}}$ is the referenced range distance.

In order to focus the signal along the azimuth dimension, the signal can be processed by the spectral analysis (SPECAN) algorithm [20,23]. According to the algorithm, the signal needs to be transformed into the time domain. In this domain, the signal can be dechirped by multiplying $\exp\left(j\pi k_{a}t^{2}\right)$. The dechirped signal is expressed as [20,23]:(8)$$\begin{array}{l} S(t,\tau )=\iint_{D}V_{d}(r_{p},t_{p})A\mathrm{sinc}\left[kT_{r}\left(\tau -\frac{2r_{p}}{cD(f_{t_{ref}})}\right)\right]W\left(t-t_{p}+\frac{f_{t_{x}}}{k_{a}}\right)\\ \;\cdot rect\left[\frac{1}{T_{b}}\left(t-t_{p}+\frac{f_{t_{c}}}{k_{a}}\right)\right]\exp\left(-j\frac{4\pi r_{p}f_{0}}{c}\right)\exp\left(-j\pi k_{a}t_{p}^{2}+j2\pi k_{a}t_{p}t\right)dr_{p}dt_{p} \end{array}$$
where
(9)$$k_{a}=\frac{2v_{r}^{2}f_{0}}{cr_{p_{ref}}}$$

The signal can then be transformed by Fourier-transform (FT) along the azimuth dimension. The transformed signal is [20,23]:(10)$$\begin{array}{l} S(t^{\prime },\tau )=\iint_{D}V_{d}(r_{p},t_{p})A^{\prime }\mathrm{sinc}\left[kT_{r}\left(\tau -\frac{2r_{p}}{cD(f_{t_{ref}})}\right)\right]W\left(T_{c}-t_{p}-t_{x}\right)\exp\left(-j\frac{4\pi r_{p}f_{0}}{c}\right)\\ \;\;\cdot \exp\left(-j\pi k_{a}t_{p}^{2}\right)\exp\left[j2\pi k_{a}(t_{p}-t^{\prime })(t_{p}-\frac{f_{t_{c}}}{k_{a}})\right]\mathrm{sinc}\left[k_{a}T_{b}\left(t^{\prime }-t_{p}\right)\right]dr_{p}dt_{p} \end{array}$$

By multiplying $\exp\left(j\pi k_{a}{t\prime }^{2}\right)$, the azimuth phase of the signal can be compensated. This is the last step of the SPECAN algorithm. At this stage, we can acquire the focused image, the expression of which can be approximated by the following equation [20,23]:(11)$$\begin{array}{l} S(t^{\prime },\tau )=\iint_{D}V_{d}(r_{p},t_{p})A^{\prime }\mathrm{sinc}\left[kT_{r}\left(\tau -\frac{2r_{p}}{cD(f_{t_{ref}})}\right)\right]W\left(T_{c}-t_{p}-t_{x}\right)\exp\left(-j\frac{4\pi r_{p}f_{0}}{c}\right)\\ \;\;\cdot \exp\left[j\pi k_{a}{\left(t^{\prime }-t_{p}^{}\right)}^{2}\right]\exp\left[-j2\pi k_{a}(t^{\prime }-t_{p})(T_{c}-t_{p})\right]\mathrm{sinc}\left[k_{a}T_{b}\left(t^{\prime }-t_{p}\right)\right]dr_{p}dt_{p} \end{array}$$

In ScanSAR interferometry, the interferometric image pair can also be expressed as Equation (11) with slow time $t_{i}$, fast time $\tau_{i}$, target time $t_{pi}$, target range $r_{pi}$, burst central time $T_{ci}$, Doppler modulation rate $k_{ai}$, burst duration $T_{bi}$ and equivalent velocity $v_{ri}$ instead of $t^{\prime }$, τ, $t_{p}$, $r_{p}$, $T_{c}$, $k_{a}$, $T_{b}$, and $v_{r}$, where the subscript “${*}_{i}$” means the image index. “*i* = 1” indicates the master image and “*i* = 2” means the slave image.

After image co-registration, the slave image can be expressed as:(12)$$\begin{array}{l} S_{2p}(t_{1},\tau_{1})=\iint_{D}V_{d}(r_{p1},t_{p1})A^{\prime }\mathrm{sinc}\left[kT_{r}\left(\tau_{1}-\frac{2r_{p1}}{cD(f_{t_{ref}})}\right)\right]\mathrm{sinc}\left[\frac{k_{a2}v_{r1}^{}}{v_{r2}^{}}T_{b2}\left(t_{1}-t_{p1}\right)\right]W\left(T_{c2}-t_{p2}-t_{x}\right)\\ \;\cdot \exp\left(-j\frac{4\pi f_{0}r_{p2}}{c}\right)\exp\left[j\pi \frac{k_{a2}v_{r1}^{2}}{v_{r2}^{2}}{\left(t_{1}-t_{p1}^{}\right)}^{2}\right]\exp\left[-j2\pi \frac{k_{a2}v_{r1}^{2}}{v_{r2}^{2}}(t_{1}-t_{p1})(\frac{v_{r2}T_{c2}}{v_{r1}}-t_{p1})\right]dr_{p1}dt_{p1} \end{array}$$

Then we substitute Equations (11) and (12) into the expression of interferometric coherence [24,25]:(13)$$\gamma (t_{1},\tau_{1})=\frac{<S_{1}(t_{1},\tau_{1})\cdot S_{2p}^{*}(t_{1},\tau_{1})>}{\sqrt{<S_{1}(t_{1},\tau_{1})\cdot S_{1}^{*}(t_{1},\tau_{1})><S_{2p}(t_{1},\tau_{1})\cdot S_{2p}^{*}(t_{1},\tau_{1})>}}$$

We can get:(14)$$\gamma (t_{1},\tau_{1})=\gamma_{a}(t_{1},\tau_{1})\cdot \gamma_{b}{\left(t_{1},\tau_{1}\right)}_{}$$
(15)$$\begin{array}{l} \gamma_{a}(t_{1},\tau_{1})=\{\int_{T}\exp\left[-j2\pi k_{a1}(t_{1}-t_{p1})(T_{c1}-\frac{v_{r2}T_{c2}}{v_{r1}})\right]\mathrm{sinc}\left[k_{a1}T_{b1}\left(t_{1}-t_{p1}\right)\right]\\ \;\cdot \mathrm{sinc}\left[k_{a1}\frac{v_{r2}T_{b2}}{v_{r1}}\left(t_{1}-t_{p1}\right)\right]dt_{p1}\}/\sqrt{\int_{T}{\mathrm{sinc}}^{2}\left[k_{a1}T_{b1}\left(t_{1}-t_{p1}\right)\right]dt_{p1}}\\ \;/\sqrt{\int_{T}{\mathrm{sinc}}^{2}\left[k_{a1}\frac{v_{r2}T_{b2}}{v_{r1}}\left(t_{1}-t_{p1}\right)\right]dt_{p1}} \end{array}$$
(16)$$\begin{array}{l} \gamma_{b}(t_{1},\tau_{1})=\left\{\int_{R}{\mathrm{sinc}}^{2}\left[kT_{r}\left(\tau_{1}-\frac{2r_{p1}}{cD(f_{t_{ref}})}\right)\right]\exp\left(-j\frac{4\pi f_{0}(r_{p1}-r_{p2})}{c}\right)dr_{p1}\right\}\;\\ \;\;/\left\{\int_{R}{\mathrm{sinc}}^{2}\left[kT_{r}\left(\tau_{1}-\frac{2r_{p1}}{cD(f_{t_{ref}})}\right)\right]dr_{p1}\right\} \end{array}$$
where $\gamma_{b}$ is the coherence caused by the baseline and $\gamma_{a}$ is the coherence caused by burst central time difference. $\gamma_{b}$ is the same as the corresponding coherence in stripmap mode, and $\gamma_{a}$ can be simplified as follows:(17)$$\gamma_{a}(t_{1},\tau_{1})=\{\begin{array}{cc} \min(T_{b1},{T^{\prime }}_{b2})/\sqrt{T_{b1}{T^{\prime }}_{b2}}, & \left|T_{c1}-{T^{\prime }}_{c2}\right|\leq \frac{\left|T_{b1}-{T^{\prime }}_{b2}\right|}{2}\\ \frac{1}{\sqrt{T_{b1}{T^{\prime }}_{b2}}}\left[\min(T_{b1},{T^{\prime }}_{b2})-\left(\left|T_{c1}-{T^{\prime }}_{c2}\right|-\frac{\left|T_{b1}-{T^{\prime }}_{b2}\right|}{2}\right)\right], & \frac{T_{b1}+{T^{\prime }}_{b2}}{2}>\left|T_{c1}-{T^{\prime }}_{c2}\right|>\frac{\left|T_{b1}-{T^{\prime }}_{b2}\right|}{2}\\ 0, & \left|T_{c1}-{T^{\prime }}_{c2}\right|\geq \frac{T_{b1}+{T^{\prime }}_{b2}}{2} \end{array}$$
where $T_{c2}^{\prime }=v_{r2}T_{c2}/v_{r1}$ and $T_{b2}^{\prime }=v_{r2}T_{b2}/v_{r1}$.

From Equation (17), we can see that the interferometric coherence is influenced by the burst central time difference $\left|\Delta T\right|=\left|T_{c1}-T_{c2}^{\prime }\right|$. The difference needs to be kept low relative to the burst duration. The interferometric coherence is also influenced by the burst duration difference as a secondary factor. The velocity difference and PRF difference relate to the burst central time difference and burst duration difference.

## 3. Increasing the Interferometric Coherence by Iterative Filtering

From the analysis of ScanSAR interferometry above, when the burst central time difference is non-negligible, the reduction of coherence should be considered. In this situation, the interferometric coherence can be increased by signal filtering.

In this method, the focused images should be transformed to the signal forms expressed in Equation (8). After that, the echoes from each target in the corresponding signal possess the same azimuth range, which facilitates the application of the filtering method. This filter can be expressed as:(18)$$f(t)=rect\left[\frac{1}{\min(T_{b1},{T^{\prime }}_{b2})-\left|\Delta T\right|+\left|T_{b1}-{T^{\prime }}_{b2}\right|/2}\left(t-\frac{T_{c1}+{T^{\prime }}_{c2}}{2}\pm \frac{\left|T_{b1}-{T^{\prime }}_{b2}\right|}{4}\right)\right]$$
where the sign ‘±’ is determined by the property of $T_{c1}-{T^{\prime }}_{c2}$ and $T_{b1}-{T^{\prime }}_{b2}$ to be positive or negative. When using this filter, the azimuth time array of the slave image should be calibrated as the time array of the master image.

Multiplying the transformed signals by this filter, the burst central times of the master and slave images become equal, leading to an increased coefficient $\gamma_{a}$.

In most cases, we do not actually know the burst central time difference, and so the filter is not precise. In this situation, iterative searches are required to find an accurate filter. The steps are shown in the following diagram (Figure 1).

In these steps, the master and slave images are first inversely processed to the signals in Equation (8). The signals can then be filtered to remove the signal parts irrelevant to interferometry. The selection of filters depends on the coherence value. We should choose the filter with the best coherence. After signal filtering, we can continue interferometric processing. The interferometric phase can then be obtained with better coherence.

During the ScanSAR signal processing, some other windows can also increase interferometric coherence (such as the Hanning window). Thus, in the iterative filtering method, a combined filter $f_{c}\left(t\right)=f\left(t\right)\cdot hanning\left(t\right)$ can be used to get better interferometric performance. In the combined filter, f(t) is the above-mentioned rectangular window, and hanning(t) is a Hanning window.

## 4. Phase Compensation of Gaofen-3 Interferometry

It was stated in Section 2 that a compensation phase $\exp\left(j\pi k_{a}{t\prime }^{2}\right)$ is required in the standard steps of ScanSAR imaging for interferometry. However, this step is not carried out because the Gaofen-3 ScanSAR images are used mainly with their amplitude information [26]. Thus, the phase $\exp\left(j\pi k_{a}{t\prime }^{2}\right)$ is not important in this situation. However, these images can still be used for interferometry if further corresponding processing is done. In this section, we analyzed the influence of this characteristic and designed a phase compensation method for the images.

In practical images, there is an unsynchronized azimuth time offset Δt between the master and slave images. After range processing and azimuth scaling for a ScanSAR echo, considering Δt, the signal in the time domain can be expressed as follows:(19)$$\begin{array}{l} S(t,\tau )=\iint_{D}V_{d}(r_{p},t_{p})A\mathrm{sinc}\left[kT_{r}\left(\tau -\frac{2r_{p}}{cD(f_{t_{ref}})}\right)\right]W\left(t-t_{p}-\Delta t+\frac{f_{t_{x}}}{k_{a}}\right)\\ \cdot rect\left[\frac{1}{T_{b}}\left(t-t_{p}-\Delta t+\frac{f_{t_{c}}}{k_{a}}\right)\right]\exp\left(-j\frac{4\pi r_{p}f_{0}}{c}\right)\exp\left[-j\pi k_{a}{\left(t-t_{p}-\Delta t\right)}^{2}\right]dr_{p}dt_{p} \end{array}$$

Multiplied by $\exp\left(j\pi k_{a}t^{2}\right)$ and transformed by FT, the focused image is:(20)$$\begin{array}{l} S(t^{\prime },\tau )=\iint_{D}V_{d}(r_{p},t_{p})A^{\prime }\mathrm{sinc}\left[kT_{r}\left(\tau -\frac{2r_{p}}{cD(f_{t_{ref}})}\right)\right]W\left(T_{c}-t_{p}-t_{x}\right)\exp\left(-j\pi k_{a}{\left(t_{p}^{}+\Delta t\right)}^{2}\right)\\ \cdot \exp\left(-j\frac{4\pi r_{p}f_{0}}{c}\right)\exp\left[j2\pi k_{a}(t_{p}+\Delta t-t^{\prime })(t_{p}+\Delta t-\frac{f_{t_{c}}}{k_{a}})\right]\mathrm{sinc}\left[k_{a}T_{b}\left(t^{\prime }-t_{p}-\Delta t\right)\right]dr_{p}dt_{p} \end{array}$$

If the image is compensated by a multiplying factor $\exp\left(j\pi k_{a}{t\prime }^{2}\right)$, we can describe the image as:(21)$$\begin{array}{l} S(t^{\prime },\tau )=\iint_{D}V_{d}(r_{p},t_{p})A^{\prime }\mathrm{sinc}\left[kT_{r}\left(\tau -\frac{2r_{p}}{cD(f_{t_{ref}})}\right)\right]W\left(T_{c}-t_{p}-t_{x}\right)\exp\left(-j\frac{4\pi r_{p}f_{0}}{c}\right)\\ \exp\left[j\pi k_{a}{\left(t^{\prime }-t_{p}^{}-\Delta t\right)}^{2}\right]\exp\left[-j2\pi k_{a}(t^{\prime }-t_{p}-\Delta t)(-\frac{f_{t_{c}}}{k_{a}})\right]\mathrm{sinc}\left[k_{a}T_{b}\left(t^{\prime }-t_{p}-\Delta t\right)\right]dr_{p}dt_{p} \end{array}$$

From this equation, we can see that the azimuth time offset Δt can be handled by azimuth shifting, and the interferometric phase will not be influenced.

However, if the phase term is not compensated, the interferometric image $S_{1}(t_{1},\tau_{1})\cdot S_{2p}^{*}(t_{1},\tau_{1})$ will have an uncompensated phase term:(22)$$P(t_{1})=exp\left(j2\pi k_{a1}t_{1}\Delta t-j\pi k_{a1}\Delta t^{2}\right)=\exp\left(j2\pi k_{a1}t_{1}\Delta t\right)\cdot \exp\left(-j\pi k_{a1}\Delta t^{2}\right)$$

This phase term is useless and will influence the interferometric phase. It can be divided into two terms: $\exp\left(-j\pi k_{a1}\Delta t^{2}\right)$ is a constant term, and only the linear phase term $\exp\left(j2\pi k_{a1}t_{1}\Delta t\right)$ is needed for compensation.

However, this compensation is not sufficient, because the velocities and PRFs are different in Gaofen-3 interferometric images. In this situation, Δt is a variant along the azimuth direction. Without considering high orders, variant Δt can be approximated as $\Delta t=\Delta t_{0}+k_{t}t_{1}$. The main term of $P(t_{1})$ then becomes $\exp\left(j2\pi k_{a1}t_{1}\Delta t_{0}\right)\cdot \exp\left(j2\pi k_{a1}k_{t}t_{1}^{2}\right)$. In the main term, $\exp\left(j2\pi k_{a1}t_{1}\Delta t_{0}\right)$ is compensated in the above-mentioned step as a linear phase term, so the second-order sub-term $\exp\left(j2\pi k_{a1}k_{t}t_{1}^{2}\right)$ should be compensated along the azimuth direction sequentially.

## 5. DEM Geolocation of Gaofen-3 Interferometry

The above sections discussed the coherence of Gaofen-3 ScanSAR interferometry, and proposed several methods to solve unsynchronized problems. Together with these discussions, the interferometric processing steps can be expressed as Figure 2.

In the processing, the interferometric images are iteratively filtered to increase their coherence. After co-registration and interferometry, an interferometric phase image can then be achieved. The phase image should be processed by flat Earth removal, phase denoising, and phase unwrapping in sequence. After phase compensation and DEM geolocation, a Gaofen-3 DEM can then be retrieved.

DEM geolocation is the last step of interferometric processing. Using the compensated unwrapped phase together with the system geometric parameters and the payload parameters, the DEM of the Earth’s surface can be extracted. This processing is based on three equations [27,28]:(23)$$v\cdot (T-S)=\lambda f_{dc}r_{s}/2$$
(24)$$\left|T-S\right|=r_{s}$$
(25)$$\left|T-S_{b}\right|=r_{s}+\lambda \phi /4\pi$$
where $v=\left(v_{x},v_{y},v_{z}\right)$ is the velocity of the satellite, $T=\left(T_{x},T_{y},T_{z}\right)$ is the position of the target, $S=\left(S_{x},S_{y},S_{z}\right)$ is the position of the satellite in the first pass, and $S_{b}=\left(S_{bx},S_{by},S_{bz}\right)$ is the position of the satellite in the second pass. These four vectors are defined in Earth-centered fixed coordinates. λ is the wave length, $f_{dc}$ is the Doppler central frequency, $r_{s}$ is the range distance, and ϕ is the interferometric phase.

By solving Equations (23)–(25), the target coordinates **T** can be obtained. One of the calculation methods able to solve the equations involves using the Newton iteration method, but this method remains time intensive. In a Gaofen-3 interferometric situation, another calculation method requires a closed-form solution to be acquired [28,29]. Because this kind of method does not use iteration, its calculation efficiency is better. According to the Gaofen-3 parameter settings, we can describe the closed-form solution as follows:(26)$$T=(c_{1x}T_{z}+c_{0x},c_{1y}T_{z}+c_{0y},T_{z}),\;T_{z}=(-c_{b}\pm \sqrt{c_{b}^{2}-4c_{a}c_{b}})/(2c_{a})$$
where the sign “±” is determined by the satellite’s looking direction. In Equation (26), the parameters can be expressed as [28]:(27)$$c_{a}=c_{1x}^{2}+c_{1y}^{2}+1,\;c_{b}=2c_{1x}c_{0x}+2c_{1y}c_{0y}-2S_{x}c_{1x}-2S_{y}c_{1y}-2S_{z},\;c_{c}=c_{0x}^{2}+c_{0y}^{2}-r_{s}^{2}-2S_{x}c_{0x}-2S_{y}c_{0y}+S\cdot S$$

The parameters $c_{0i}$ and $c_{1i}$ (*i* = *x*, *y*) in the above equations are expressed as:(28)$$c_{0i}=m_{0i}r_{1}+m_{1i}r_{2},\;c_{1i}=-m_{0i}v_{z}-m_{1i}(S_{z}-S_{bz})$$
where
(29)$$M=\left[\begin{array}{cc} m_{0x} & m_{1x}\\ m_{0y} & m_{1y} \end{array}\right]={\left[\begin{array}{cc} v_{x} & v_{y}\\ S_{x}-S_{bx} & S_{y}-S_{by} \end{array}\right]}^{-1}$$
(30)$$r_{1}=\lambda f_{dc}r_{s}/2+v\cdot S,\;r_{2}=\{S\cdot S-S_{b}\cdot S_{b}+{\left[\lambda \phi /(4\pi )\right]}^{2}+\lambda \phi r_{s}/(2\pi )\}/2$$

During processing, the absolute interferometric phase ϕ is required. However, from the compensated unwrapped phase, only the relative phase can be achieved. A system phase $\phi_{0}$ should be compensated to the relative phase. We use GCPs to determine the phase $\phi_{0}$. The point heights can be derived from known DEM data, such as SRTM DEM. The height of a GCP can be expressed as:(31)|T|=h

Combining and solving Equations (23), (24), and (31), we can find coordinates T of a GCP. The closed-form solution of the equations is the same as Equation (26), except that some parameters should be replaced:(32)$$M=\left[\begin{array}{cc} m_{0x} & m_{1x}\\ m_{0y} & m_{1y} \end{array}\right]={\left[\begin{array}{cc} v_{x} & v_{y}\\ S_{x} & S_{y} \end{array}\right]}^{-1}$$
(33)$$r_{2}=(S\cdot S+h^{2}-r_{s}^{2})/2$$

Substituting the GCP coordinates into Equation (25), we can find the absolute interferometric phase ϕ of a GCP. Subtracting the relative interferometric phase from ϕ, phase $\phi_{0}$ can be obtained. Phases $\phi_{0}$ from multiple GCPs can be then averaged. We can then acquire the absolute interferometric phase image of all the points by compensating the average phase $\phi_{0}$.

In the above method, system errors are not considered. In presence of some system errors, the system phase $\phi_{0}$ varies along the range and azimuth directions, and it can be expressed as $\phi_{e}\left(t_{1},r_{1}\right)$. From the system phases of GCPs, some system errors can be estimated and then compensated for, including the azimuth phase error discussed in Section 4. Thus, the phase compensation discussed in Section 4 can be combined with the DEM geolocation processing.

The system phase $\phi_{e}\left(t_{1},r_{1}\right)$ can be expressed as:(34)$$\phi_{e}(t_{1},r_{1})=\phi_{0}+k_{ae1}t_{1}+k_{ae2}t_{1}^{2}+k_{re1}r_{1}+k_{c1}t_{1}r_{1}+k_{c2}t_{1}^{2}r_{1}\;k_{ae1}=2\pi k_{a1}\Delta t_{0}+k_{ab1},\;k_{ae2}=2\pi k_{a1}k_{t}+k_{ab2}$$
where $k_{re1},\;k_{ab1},\;k_{ab2},k_{c1},\;\mathrm{and}\;k_{c2}$ are phase error coefficients caused by baseline error.

If we obtain the system phase values of multiple GCPs, we can estimate the compensation coefficients using the least square method:(35)$$K={\left(P^{\prime }P\right)}^{-1}P^{\prime }\Phi_{e}$$
(36)$$K=(\phi_{0},k_{ae1},k_{ae2},k_{re1},k_{c1},k_{c2}{)}^{\prime },\;\Phi_{e}=(\phi_{e1},\cdots ,\phi_{ei},\cdots ,\phi_{eN}{)}^{\prime }$$
(37)$$P=\left(\begin{array}{cccccc} 1 & t_{11} & t_{11}^{2} & r_{11} & t_{11}r_{11} & t_{11}^{2}r_{11}\\ \cdots & \cdots & \cdots & \cdots & \cdots & \cdots \\ 1 & t_{1i} & t_{1i}^{2} & r_{1i} & t_{1i}r_{1i} & t_{1i}^{2}r_{1i}\\ \cdots & \cdots & \cdots & \cdots & \cdots & \cdots \\ 1 & t_{1N} & t_{1N}^{2} & r_{1N} & t_{1N}r_{1N} & t_{1N}^{2}r_{1N} \end{array}\right)$$
where the subscript “**_i_* ” means the GCP index, and “N” is the number of GCPs.

With the estimated compensation coefficients, we can calculate the system phase $\phi_{e}$ of each master image pixel according to Equation (34). The compensated phase image can then be acquired by compensating phase $\phi_{e}$; at the same time the influence of azimuth phase error and baseline error can be weakened.

Based on the above discussions, GCPs are used to acquire absolute phase and compensate phase error. Because the GCP data are their three-dimensional coordinates in the geodetic coordinates system, we still need to find the positions of the GCPs in the phase image before the above geolocation processing. First, the GCP coordinates should be transformed from the geodetic coordinates to Earth-centered fixed coordinates. Then, for each GCP, its azimuth time $t_{p}$ and range distance $r_{s}$ need to be calculated. These two parameters can determine the position of each GCP in the phase image.

In the calculation of a GCP’s $t_{p}$ and $r_{s}$, the corresponding satellite position can be approximated as $S=S_{0}+v_{0}t_{p}$, where $S_{0}$ and $v_{0}$ are the satellite position and velocity at the reference time $t_{0}$. Thus, we can calculate the azimuth time $t_{p}$ as follows:(38)$$t_{p}=(-p_{b}\pm \sqrt{p_{b}^{2}-4p_{a}p_{c}})/(2p_{a})$$
(39)$$p_{a}={\left|v_{0}\right|}^{2}-4{\left|v_{0}\right|}^{4}/(\lambda^{2}f_{dc}^{2}),\;p_{b}=-2\cdot v_{0}\cdot (T_{p}-S_{0})+8{\left|v_{0}\right|}^{2}\cdot v_{0}\cdot (T_{p}-S_{0})/(\lambda^{2}f_{dc}^{2}),\;p_{c}={\left|T_{p}-S_{0}\right|}^{2}-4\cdot {\left[v_{0}\cdot (T_{p}-S_{0})\right]}^{2}/(\lambda^{2}f_{dc}^{2})$$
where the sign “±” is determined by the squint angle of a GCP and $T_{p}$ is the coordinates of the GCP.

The approximation “$S=S_{0}+v_{0}t_{p}$” does not consider the velocity variation. In order to decrease this influence, we must make a new approximation as $S=S_{02}+v_{02}t_{p2}$, where $S_{02}$ and $v_{02}$ are the actual satellite position and velocity at the reference time $t_{0}+t_{p}$. We repeat the calculation as Equations (38) and (39), and a new azimuth time $t_{p2}$ can thus be obtained. With the same method, we can acquire a third new azimuth time $t_{p3}$. Thus, the final azimuth time “$t_{pf}=t_{p}+t_{p2}+t_{p3}$”, which refers to $t_{0}$, can be determined. Range distance $r_{s}$ at the azimuth time $t_{pf}$ can be calculated with Equation (24). Thus, the GCP can be located in the phase image. From the GCP coordinates, we can obtain the approximate height of the nearest grid point. The above geolocation and compensation can then be carried out.

## 6. Results and Discussion

The above sections analyzed the coherence of ScanSAR interferometry and studied several problems in Gaofen-3 processing. In this section, we carried out a simulation and practical interferometric processing to explain the analysis and processing methods. For interferometric processing, we used the above-mentioned Gaofen-3 interferometric images over Kunlun Mountain. From the interferometric processing, the iterative filtering method, phase compensation, and DEM geolocation were verified.

### 6.1. Iterative Filtering Method

In Section 2, we discussed the interferometric performance related to the burst central time difference and burst duration difference. The relationship between the burst central time, burst duration difference, and the coherence is shown in Figure 3.

In Figure 3, “*k*” is a coefficient and $k=T_{b2}^{\prime }/T_{b1}$, which means the ratio of burst durations. We express the ratio of burst central time difference to shorter burst duration as “*k_c_*”. Considering *k* = 1, the coherence is only influenced by the burst central time difference. In this situation, if $k_{c}=0$, the coherence is not influenced. With an increase of $k_{c}$, the coherence decreases linearly. When $k_{c}$ exceeds 1—that is to say, when the burst central time difference exceeds the burst duration—the coherence is reduced to 0. The interferometric processing will fail in this decorrelation situation. Considering *k* = 1.2, the coherence will be influenced by burst duration difference. In this situation, when $k_{c}$ is within 0–0.1, the coherence value is 0.91 and is mostly lower than that in “*k* = 1” situation. When $k_{c}$ is from 0.1 to 1.1, the coherence decreases linearly, but it is better than that in the *k* = 1 situation. When $k_{c}$ exceeds 1.1, the coherence reduces to 0. For the images in this paper, *k* was near 1.004. Thus, the coherence of these images was mainly influenced by the burst central time difference.

In Gaofen-3 interferometric images, it is difficult to maintain a zero burst central time difference. As a consequence, interferometric coherence will be more or less influenced. When the burst central time difference is relatively large, the iterative filtering method described in Section 3 can be used to alleviate the influence.

Two interferometric images, shown in Figure 4, were used to verify the filtering method. These two interferometric images were cut from the Kunlun Mountain images with a relatively big burst central time difference.

Filtering the two images with different burst central time differences |ΔT|, we found different coherence values after interferometry. This coherence was estimated from the interferometric images. |ΔT| versus coherence is shown in Figure 5.

From Figure 5, when we used the rectangular filters, and |ΔT| used in the filters reached 70 pixels, the interferometric coherence increased by 0.05. When we used the combined filters, and |ΔT| used in the filters reached 70 pixels, the interferometric coherence increased by 0.02. With these two kinds of filters, the best |ΔT| values were all 70 pixels. With the rectangular filters, the decorrelation caused by the burst central difference was 70/582 = 0.12, and the coherence caused by other factors was 0.55, where 582 was the azimuth band sample. Thus, the coherence increases by 0.12×0.55=0.066 theoretically, and the experiment result of 0.05 was close to the theoretical value. From Figure 5b, when the burst central time difference used in the combined filters was 0 pixels, the coherence was better than that of the value shown in Figure 5a. This is because the Hanning window decreased the amplitude of the unsynchronized signal part. When the burst central time difference used in the combined filters was 70 pixels, the coherence was better than that of the value shown in Figure 5a. This means that the Hanning window increased the coherence. Thus, it was suitable to use combined filters in the iterative filtering method.

### 6.2. Phase Compensation

In Gaofen-3 ScanSAR interferometry, as discussed in Section 4, a linear phase term and a second-order phase term should be compensated along the azimuth direction. By first applying this compensation method with a linear phase term on a pair of Gaofen-3 interferometric images (Figure 6a,b), we can get a compensated interferometric phase, as shown in Figure 6. These images were also cut from the Kunlun Mountain images.

After interferometry, the original denoised interferometric phase is shown in Figure 6c. Figure 6d shows the compensated denoised interferometric phase, Figure 6e shows the original unwrapped phase after flat Earth removal along the range direction, and Figure 6f shows the corresponding compensated phase. From these figures, we can see that the compensation solved the phase’s linear slope along azimuth direction.

In the above figures, the velocity of the master image was 7.5674 km/s and its PRF was 1185.6 Hz, while the velocity of the slave image was 7.5679 km/s and its PRF was 1190.4 Hz. As discussed in Section 4, these differences resulted in a second-order term along the azimuth direction. Compensating the Gaofen-3 interferometric phase with a second-order term, we obtained the following results.

In these figures, Figure 7a shows the second-order compensated denoised interferometric phase, and Figure 7b shows the second-order compensated unwrapped phase after flat Earth removal along the range direction. From the results, second-order compensation was able to solve the phase curving effect along the azimuth direction.

In the interferometric phase, we found periodic lines. These lines were located at the areas where different bursts intersected. The burst central time difference in these areas neared the burst cycle time. Thus, based on the discussion in Section 2, the coherence in these areas was 0 and normal interferometric phase stripes could not be formed. This influence can be overcome by bursts aligned between the master and slave images before ScanSAR burst splicing. This aligning method is the best method. However, if we cannot obtain the interferometric images before burst splicing, the interpolation method can be used to fill in the invalid areas.

### 6.3. DEM Geolocation

From the above processing, a compensated unwrapped interferometric phase image was achieved. Subsequently, the satellite position and velocity during the observing time, as well as the Doppler central frequency and the target range distance were obtained from the Gaofen-3 information file. We then chose several GCPs in the master image. GCP height information can be obtained from a known DEM. According to the method in Section 5, we obtained the DEM of the tested Earth’s surface as follows.

In Figure 8, Figure 8a shows the Gaofen-3 DEM, with imaging coordinates covering a 9 km (range) × 20 km (azimuth) area, and Figure 8b shows the top view of the Gaofen-3 DEM. The geographical characteristics of the DEM were coincident with those of the master image. Compared with the SRTM data of the same area (Figure 9), the achieved DEM matched the SRTM DEM (a 30 m × 30 m grid) [30].

In order to evaluate the Gaofen-3 DEM quantitatively, we chose 10 check points from the SRTM DEM, marked with “+” in Figure 9. The height comparisons of the Gaofen-3 DEM and the SRTM DEM for these check points are listed in Table 2.

As seen in Figure 8 and Figure 9, the Gaofen-3 DEM was coarser than the SRTM DEM. As shown in Table 2, the average height precision of the Gaofen-3 DEM was about 25 m, and the maximum height error of the check points reached 44 m (absolute value). Height errors of the SRTM DEM samples were lower than 16 m. These results occurred because of the differences between the Gaofen-3 and SRTM interferometry. The Gaofen-3 DEM was acquired in ScanSAR mode and its grid was about 160 × 160 m, while the SRTM DEM was acquired in stripmap mode, and its grid was about 30 × 30 m; Gaofen-3 has a coarser grid. Gaofen-3 features repeat-pass interferometry and SRTM uses single-pass interferometry, so the coherence of Gaofen-3 should theoretically be lower than that of SRTM. Consequently, the Gaofen-3 DEM’s quality was in accord with Gaofen-3′s system characteristics. As the geographical characteristics of these DEMs were consistent, the accuracy of the Gaofen-3 DEM was verified.

By applying the above-mentioned interferometric processing to a wide area, we obtained the Gaofen-3 DEM as Figure 10.

The produced DEM was also of Kunlun Mountain, covering a 70 (range) × 35 km (azimuth) area. ScanSAR interferometry is suitable for this kind of wide-area mapping. Further, wide-area mapping can be dealt with by block processing and splicing, and the above 70 × 35 km area can be treated as a block.

## 7. Conclusions

This paper discussed interferometric analyzing and processing methods for Gaofen-3 images in ScanSAR mode. The conditions for ScanSAR interferometry are more rigorous than those of normal stripmap SAR interferometry. We analyzed the coherence in ScanSAR interferometry in detail to determine these conditions. From the analysis, the burst central time difference between the master and slave images was shown the coherence. In order to reduce the influence, we presented an iterative filtering method able to remove the signal parts irrelevant to interferometry, so as to increase the coherence. The analysis and the filtering method can also be influenced by burst duration difference and velocity difference, which should be incorporated in the analysis and filters. In Gaofen-3 ScanSAR interferometry, the phase error along the azimuth direction is severe. We analyzed the cause of the phase error, and correspondingly proposed a linear phase compensation and a second-order phase compensation to determine the right interferometric phase. In the DEM geolocation of Gaofen-3 interferometry, we derived a closed-form solution with GCP information. Without complex iteration in the method, a closed-form solution was able to efficiently retrieve a DEM of the Earth’s surface. These methods were applied to Gaofen-3 ScanSAR images and returned good results. These methods could also help to realize ScanSAR interferometry for other similar satellites.

## Figures and Tables

**Figure 1 sensors-19-04689-f001:**
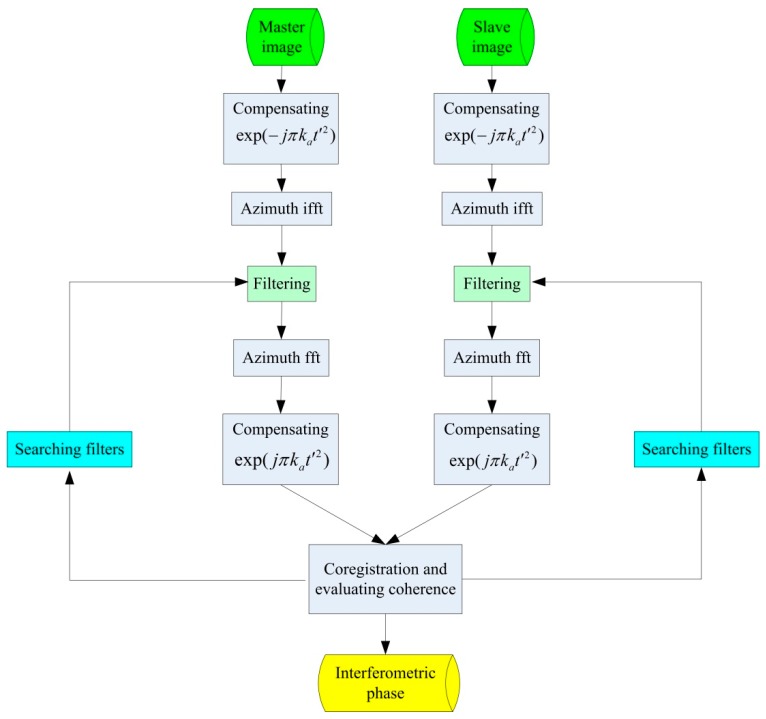
The iterative filtering method diagram.

**Figure 2 sensors-19-04689-f002:**
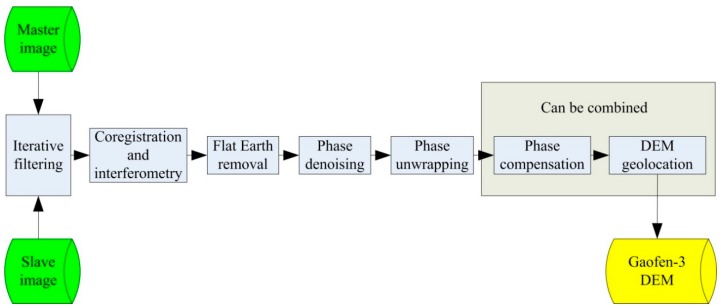
The interfeometric processing diagram. DEM: digital elevation model.

**Figure 3 sensors-19-04689-f003:**
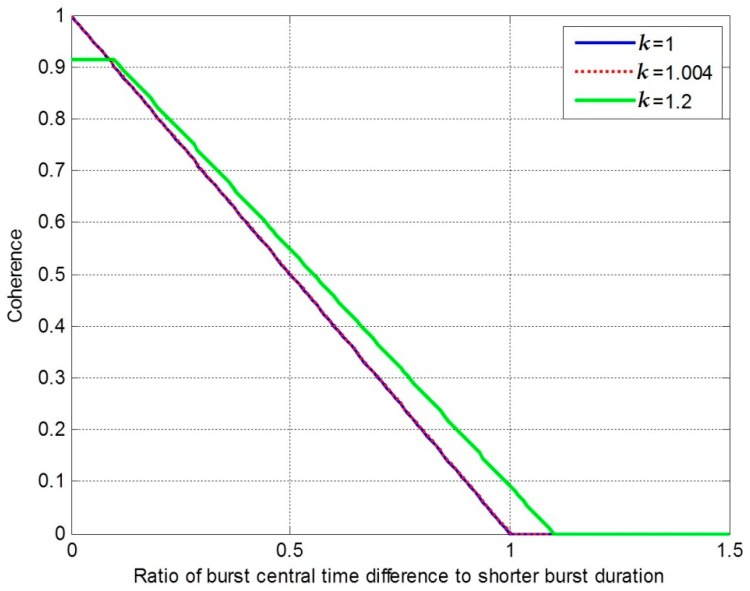
The relationship between the ratio of burst central time difference to shorter burst duration and coherence.

**Figure 4 sensors-19-04689-f004:**
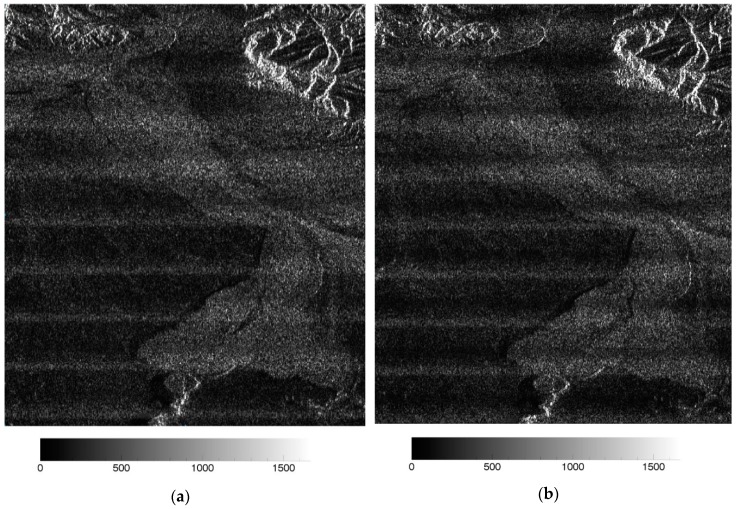
Gaofen-3 SAR images. (**a**) The master image; (**b**) the slave image.

**Figure 5 sensors-19-04689-f005:**
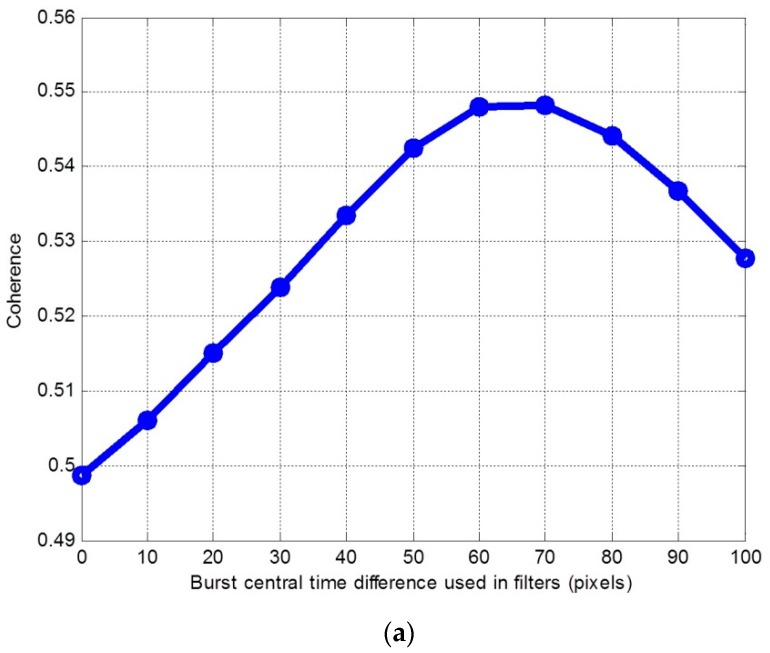
|ΔT| versus coherence processed from the two images shown in Figure 4. (**a**) Results when using rectangular filters; (**b**) results when using combined filters.

**Figure 6 sensors-19-04689-f006:**
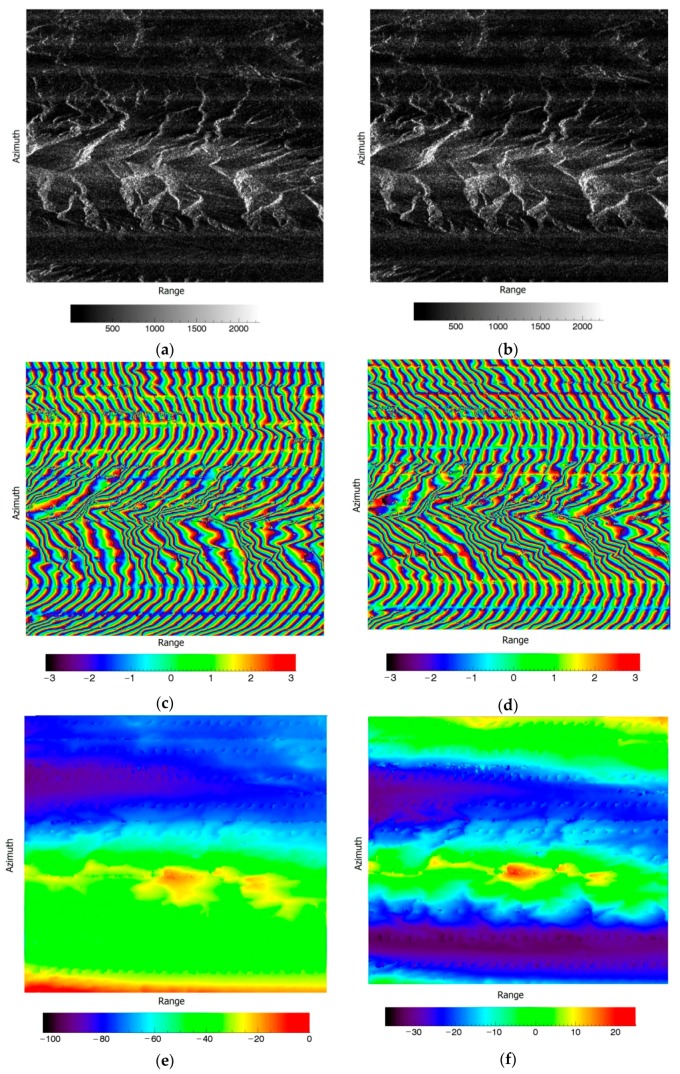
The linear compensation results. (**a**) The master image; (**b**) the slave image; (**c**) the original denoised interferometric phase (rad); (**d**) the compensated denoised interferometric phase (rad); (**e**) the original unwrapped phase after flat Earth removal along the range direction (rad); (**f**) the compensated unwrapped phase after flat Earth removal along the range direction (rad).

**Figure 7 sensors-19-04689-f007:**
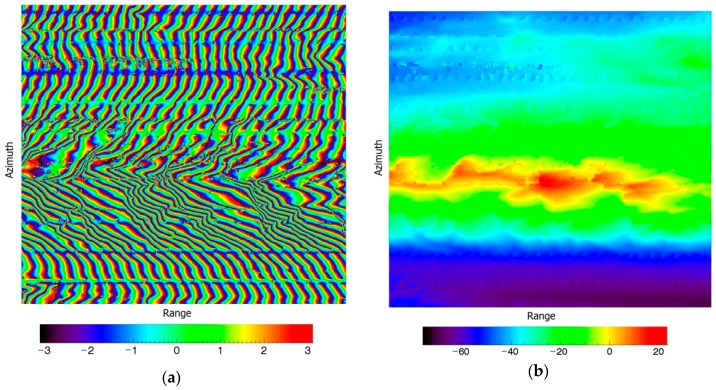
The second-order compensation results. (**a**) The second-order compensated denoised interferometric phase (rad); (**b**) the second-order compensated unwrapped phase after flat Earth removal along the range direction (rad).

**Figure 8 sensors-19-04689-f008:**
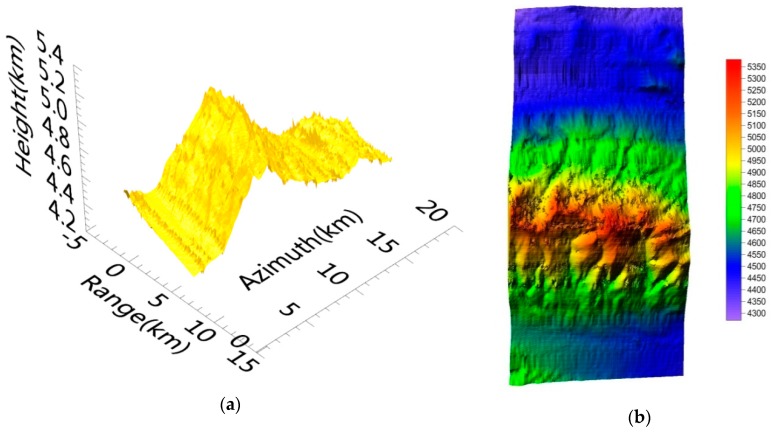
Gaofen-3 DEM after geolocation. (**a**) Gaofen-3 DEM in imaging coordinates; (**b**) top view of the Gaofen-3 DEM (m).

**Figure 9 sensors-19-04689-f009:**
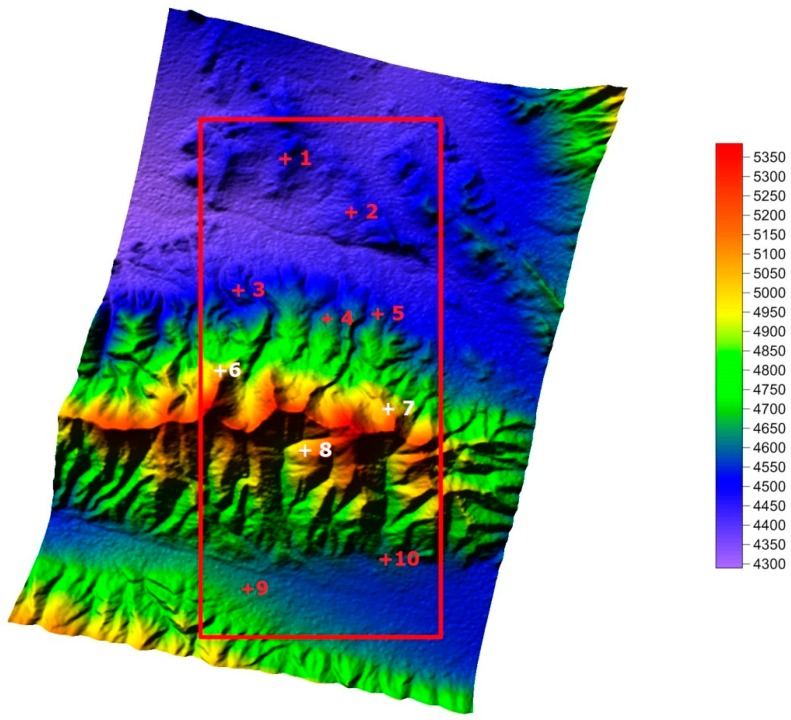
Shuttle Radar Topography mission (SRTM) DEM of the same area (m).

**Figure 10 sensors-19-04689-f010:**
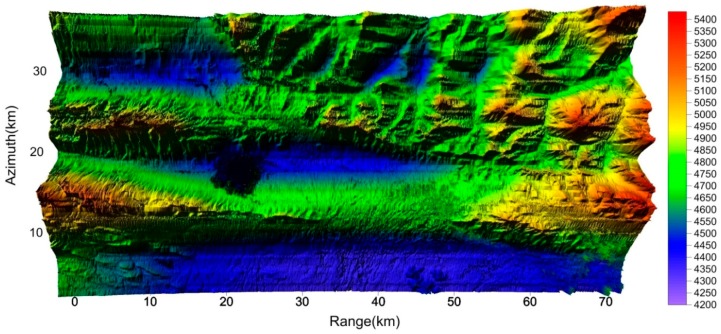
Gaofen-3 DEM covering a 70 km (range) × 35 km (azimuth) area (m).

**Table 1 sensors-19-04689-t001:** Main parameters of the Gaofen-3 interferometric images. PRF: pulse repetition frequency.

Parameters	Master Image	Slave Image
Central frequency (GHz)	5.4	5.4
Center look angle (°)	34.7	34.7
PRF (Hz)	1185.637085	1190.421753
Satellite velocity (km/s)	7.5674	7.5679
Band width (MHz)	30	30
Pulse width (μs)	45	45
Pulse number	100	100

**Table 2 sensors-19-04689-t002:** Height comparisons of the Gaofen-3 DEM and the SRTM DEM for check points.

Index	1	2	3	4	5	6	7	8	9	10
Gaofen-3 DEM (m)	4496	4466	4489	4593	4584	5090	5061	4911	4683	4643
SRTM DEM (m)	4472	4454	4533	4630	4597	5050	5029	4923	4657	4633
Height difference (m)	24	12	−44	−37	−13	40	32	−12	26	10

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
