# Peer review of "ScanSAR Interferometry of the Gaofen-3 Satellite with Unsynchronized Repeat-Pass Images"

_sensors, 2019, doi:10.3390/s19214689_

Round 1

Reviewer 1 Report

This paper describes a technique to improve the interferometric coherence
from Gaofen-3 ScanSAR images. ScanSAR is getting popular for its wide coverage. This will be an interesting paper for the readers. There is one general comment and some detailed comments will be listed afterwards.
The title of the paper is too general. I suggest that it should be modified to
include few words regarding the technique mentioned in the paper.

(Ln35) Please give some more background of Gaofen-3 satellite spec such as revisiting days, resolution, ... etc.
(Ln40) Replace "thought about" by "considers"
(Ln41) "two interferometric images" ?? one interferometric image comes from a pair of  SAR images. Are you saying two pairs of SAR images ? Please clarify this part.
(Ln53) Replace ". Its used SAR in ScanSAR mode could"by "which is capable to"
(Ln222) How does Gaofen-3 performe in burst central time difference compared to  the satellite in the same band? (e.g. Sentinel-1A or Radarsat)
(Ln227) I think it is more clear to state "SAR images" than "interferometric images". interferometric image comes from a pair of SAR images. Also, please include the colorbar.
(Ln233) I observe there is an optimal |deltaT| to improve the coherence. Although the method is applied, it seems you still need to figure out the best |deltaT|. I think the sentence starting "We can..." does not follow the logic of its previous sentence. Please rephrase it to be more clear and consistent.
(Ln241) Figure 5 Please include the coordinates, supposed to be azimuth and range ? Also please include colorbar.
(Ln249) Replace "slop" by "slope"
(Ln279) Figure 8, I think it is better to provide a difference map between yours and SRTM. Otherwise, there is not much information in comparing two similar map, saying figure 8 and figure7(b).

Author Response

Response to Reviewer 1 Comments

Point 1: The title of the paper is too general. I suggest that it should be modified to include few words regarding the technique mentioned in the paper. 

Response 1: I have changed the title from “Gaofen-3 Satellite SAR Interferometry in ScanSAR Mode” to “ScanSAR Interferometry of the Gaofen-3 Satellite with Unsynchronized Repeat-pass Images”. In the new title, I show that my main works in the paper are to deal with the unsynchronized characteristics.

Point 2: (Ln35) Please give some more background of Gaofen-3 satellite spec such as revisiting days, resolution, ... etc.

Response 2: I have revised this part. In the revision, I have given orbit, altitude and revisiting period of Gaofen-3. According to different working modes, I have listed their resolutions and swathes.

Point 3: (Ln40) Replace "thought about" by "considers".

Response 3: I have replaced  "thought about" by "considers".

Point 4: (Ln41) "two interferometric images" ?? one interferometric image comes from a pair of  SAR images. Are you saying two pairs of SAR images ? Please clarify this part.

Response 4: In this sentence, I am saying  two SAR images. In the two images, an image is achieved after another one with an interval of 29 days in the repeat-pas mode.  I have revised it to “two SAR images”.

Point 5: (Ln53) Replace ". Its used SAR in ScanSAR mode could"by "which is capable to".

Response 5: I have replaced  ". Its used SAR in ScanSAR mode could" by "which is capable to".

Point 6: (Ln222) How does Gaofen-3 performe in burst central time difference compared to  the satellite in the same band? (e.g. Sentinel-1A or Radarsat).

Response 6: Compared to Radarsat, the burst central time difference of Gaofen-3 varies along azimuth direction because of velocity difference, burst duration difference and PRF difference. So the influence of burst central time difference varies along azimuth direction. In part of image, the influence can be very severe.

Point 7: (Ln227) I think it is more clear to state "SAR images" than "interferometric images". interferometric image comes from a pair of SAR images. Also, please include the colorbar.

Response 7: I have replaced  " interferometric images" by " SAR images ". I have added the colorbar.

Point 8: (Ln233) I observe there is an optimal |deltaT| to improve the coherence. Although the method is applied, it seems you still need to figure out the best |deltaT|. I think the sentence starting "We can..." does not follow the logic of its previous sentence. Please rephrase it to be more clear and consistent.

Response 8: I have rewritten the sentences in this part in order to describe the experiment results more clearly and consistently. In the revision, I have figure out the best  |deltaT|.

Point 9: (Ln241)Figure 5 Please include the coordinates, supposed to be azimuth and range ? Also please include colorbar.

Response 9: I have added coordinates and colorbar to Figure 5 (Figure 6 in the revision).

Point 10: (Ln249)Replace "slop" by "slope"

Response 10: I have replaced "slop" by "slope".

Point 11: (Ln279) Figure 8, I think it is better to provide a difference map between yours and SRTM. Otherwise, there is not much information in comparing two similar map, saying figure 8 and figure7(b).

Response 11: In the revision, I have chosen 10 check points in the DEM to compare the difference between mine and SRTM data. I think this comparison can reflect the DEM difference quantitatively.

Thank you very much!

Reviewer 2 Report

This manuscript uses Gaofen-3 Satellite ScanSAR mode data for interferometry processing. However, the innovative is limiited and the result can not be accted in current form, and there is a lack of verification of the generality of the algorithm. Currently, this manuscript is not ready for publication.

The burst unsynchronization in ScanSAR interferometry leads to the decrease of interferogram coherence. This is because the azimuth spectrum corresponding to the same target in the burst of master and slave images does not overlap. The factors mentioned in the manuscript such as burst central time difference, burst duration difference and velocity difference are all responsible for the non-overlapping of the azimuth spectrum. The method used in the manuscript to solve the ScanSAR burst unsynchronization are very classic and commonly used. The processing results in the manuscript are not satisfactory. Using the method proposed in the paper, the coherence coefficient is only increased by 0.02, which is too small. The entire interferometry process description is not clear. And there is no quantitative comparison of the results obtained. The generality of the algorithm indicates a deficiency. Only a small area was selected for display, which does not explain the ability of the Gaofen-3 satellite in ScanSAR interferometry. Moreover, the scene information is not presented. The fact that Gaofen-3 satellite can perform interferometry processing has been demonstrated in some literature. Therefore, the meaning of this manuscript is not significant.

Author Response

Response to Reviewer 2 Comments

Point 1: The burst unsynchronization in ScanSAR interferometry leads to the decrease of interferogram coherence. This is because the azimuth spectrum corresponding to the same target in the burst of master and slave images does not overlap. The factors mentioned in the manuscript such as burst central time difference, burst duration difference and velocity difference are all responsible for the non-overlapping of the azimuth spectrum. The method used in the manuscript to solve the ScanSAR burst unsynchronization are very classic and commonly used. The processing results in the manuscript are not satisfactory.

Response 1: Really my paper is not an intense innovation. But I think my paper still has some moderate innovations. For interferometric coherence analysis, I have thought about more complete unsynchronized factors, including differences of burst central times, burst durations, PRFs, velocities. This analysis can help reapeat-pass InSAR to optimize its working parameters. I analysed the influence of unsynchronized time offset on azimuth phase error and compensated the phase error. I have revised the DEM geolocation method to integrate the azimuth phase compensation. So compensating coefficients can be estimated from GCPs. I have also revised the iterative method by considering other windows. 

    The unsatisfactory processing results are mainly because of the image characteristics. The coherence of the used image pair is low, so I cannot obtain high precision DEM. Because of burst unsynchronization, there are periodic decorrelated lines in phase image. So the DEM exists unsatisfactory lines. My processing methods have solved some problems. But some other problems can only be dealt by InSAR system optimization. I think processing of relative unsatisfactory data is still meaningful. We can find influential factors and help InSAR optimization.

Point 2: Using the method proposed in the paper, the coherence coefficient is only increased by 0.02, which is too small.

Response 2: I have revised this experiment. The result 0.02 is achieved when using combined filters, the Hanning windows in the combined filters can reduce the coherence increase. With rectangular filters, the coherence can increase 0.05, which is 10% of the coherence value. If the interferometric image pair has better coherence and more burst central time difference, the coherence increase will be more significant.

Point 3: The entire interferometry process description is not clear.

Response 3: At the beginning of Section 5, I have added the block diagram of interferometric data processing. The block diagram introduces the interferometric processing steps. At the same time, I have added the interferometry process description in this place.

Point 4: There is no quantitative comparison of the results obtained.

Response 4: In the revision, I have chosen 10 check points in the DEM to compare the difference between mine and SRTM data. I think this comparison can reflect the DEM difference quantitatively.

Point 5: The generality of the algorithm indicates a deficiency.

Response 5: I have revised my algorithm and experiment descriptions to make them more clearly. I think the problems in the paper will also be met by other SAR’s ScanSAR interferometry, and my algorithms are derived from general models.

Point 6: Only a small area was selected for display, which does not explain the ability of the Gaofen-3 satellite in ScanSAR interferometry. Moreover, the scene information is not presented.

Response 6: In the revision, I have added Table 1 to present the scene information. I have also added Figure 10 to explain the ability of the Gaofen-3 satellite in ScanSAR interferometry. The DEM in Figure 10 covers a 70km(Range) ×35km(Azimuth) wide area.

Point 7: The fact that Gaofen-3 satellite can perform interferometry processing has been demonstrated in some literature. Therefore, the meaning of this manuscript is not significant.

Response 7: This paper discusses the Gaofen-3 interferometry in the ScanSAR mode. The main purpose is to solve several problems related to the characteristics of  ScanSAR interferometry. Because the ScanSAR mode is a burst mode, its interferometric performance and processing will be influenced by burst unsynchronizations. Yet other modes such as stripmap mode are not influence by these problems. The Gaofen-3 interferometry in some literatures are mainly for stripmap mode, not for ScanSAR mode. So I think this paper is mainly different from some other Gaofen-3 interferometry literatures. 

Thank you very much!

Reviewer 3 Report

Dear Editor, Dear Authors

This manuscript proposed an iterative filtering phase compensation method to process Gaofen-3 ScanSAR interferometry. They analyzed the coherence lose due to burst timing shifts, and try to correct the phase errors along the azimuth direction. The applied their method to form a Gaofen-3 ScanSAR interferogram, and converted the phase to height.

In my opinion, this work did not reach the quality for a publication at Sensors, as:

Given the resolution and baseline configuration, DEM generation is not a good target application for ScanSAR images. The topic will have very limited interest for the SAR application community and for readers of Sensors. Section 2, equations are mostly from published work [20]-[22]. Line 232, the coherence increase is only 0.02, very limit improvement for such efforts. As Authors stated that the Gaofen3 ScanSAR mode is not designed for SAR interferometry. I’m not sure why we need to spend the large processing step to get a coarse DEM, which is available anyway from SRTM DEM. From the results presented in Figure 5 and 6. The phase discontinuities are still apparent. I highly doubt if the present method is useful for a real application. Again, from the result presented in Figure 7, the burst stich is not acceptable at all, making the application useless. English needs to be significantly improved. Many sentences are vague and wording is not precise.

I’m sorry I cannot give more positive conclusion this time.

Regards

Author Response

Response to Reviewer 3 Comments

Point 1: Given the resolution and baseline configuration, DEM generation is not a good target application for ScanSAR images. The topic will have very limited interest for the SAR application community and for readers of Sensors.

Response 1: I think ScanSAR interferometry still has its own advantages. Really its resolution is lower than other SAR modes, but it has wide swath. So ScanSAR interferometry can map a wide area in very short time. The influence of lower resolution is the coarser grid of produced DEM, DEM height information can still keep good if the interferometric SAR system are well optimized. In some applications, we need rapid mapping of wide area rather than precise grid. ScanSAR interferometry is also suitable to be combined with DInSAR technology to survey the earthquake and ground subsidence of a wide area with coarse grid.

Point 2: Section 2, equations are mostly from published work [20]-[22].

Response 2: In Section 2, I have revised my paper to indicate the equations referred to references [20]-[22] more clearly. I have listed these equations here because they are the basis of my coherence analysis and azimuth phase error analysis.  Further I have made some improvement for the referenced equations in order to apply the models to my analyses.

Point3: Line 232, the coherence increase is only 0.02, very limit improvement for such efforts.

Response 3: I have revised this experiment. The result 0.02 is achieved when using combined filters, the Hanning windows in the combined filters can reduce the coherence increase. With rectangular filters, the coherence can increase 0.05, which is 10% of the coherence value. If the interferometric image pair has better coherence and more burst central time difference, the coherence increase will be more significant.

Point 4: As Authors stated that the Gaofen3 ScanSAR mode is not designed for SAR interferometry. I’m not sure why we need to spend the large processing step to get a coarse DEM, which is available anyway from SRTM DEM.

Response 4: In the revision, I have revised this sentence. The meaning of this sentence is that Gaofen-3 ScanSAR images are used mainly with their amplitude information, so the compensation phase  is not thought about in the images. But these images can still be used for interferometry if further corresponding processing is made. 

Because the ground topography is changeable, the ground DEM needs to be remapped after a period of time. In some applications, we need rapid mapping of wide area rather than precise grid.  In surveying the earthquake and ground subsidence of a wide area, we can use DEM difference to monitor the ground deformation. In this situation, only coarse grid DEM deformation is needed.

Point 5: From the results presented in Figure 5 and 6. The phase discontinuities are still apparent. I highly doubt if the present method is useful for a real application. Again, from the result presented in Figure 7, the burst stich is not acceptable at all, making the application useless. 

Response 5: This problem is because of burst unsynchronization. There are periodic decorrelated lines in the phase image, so the DEM has unsatisfactory lines too. This problem can be overcome by aligning bursts between the master and slave images before splicing the ScanSAR burst images. Because the images that I can acquire are images after burst image splicing, this problem cannot be solved by interferometric processing. This problem assuredly can influence the DEM quality, but the presented method is not about this problem, the experiment of the present method will not be influenced by the problem. I think processing of relative unsatisfactory data is still meaningful. We can find influential factors and help InSAR optimization.

Point 6: English needs to be significantly improved. Many sentences are vague and wording is not precise.

Response 6: In the revision, I have submitted my paper to the MDPI English editing service. They have revised my paper in detail.

Thank you very much!

Reviewer 4 Report

Paper is interesting, I recommend publishing it in Remote sensing journal. Some of the sensor properties have already been published in Sensors. Paper needs major revision. Comments are the following:

Satellite data are missing. It would be useful to include a table with satellites parameters. Authors stated that images are acquired using different PRF, velocities, burst duration, but values are not mentioned at the beginning of the paper. They are mentioned in Experimental results, but they belong in section 2. For specialist it is clear what is SCAN SAR mode, for no specialist some introduction is missing and goals and advantages/disadvantages should be discussed. Block diagram of SAR data processing is missing, with focusing algorithm. In section 3 authors compensated the interferometric coherence by iterative filtering. What is a gain or increase in the coherence? 0.02 for image shown in Fig. 3? Rect. windows are usually the worst one to use. Have you considered to use any other filter windows? Could you include atmospheric corrections in the model? Page 9. Longer burst duration, average burst duration, burst central time, all data are missing. 5. what is a ground truth? How well the interferometric information fits or describe real scene? A comparison with other sensors is missing?

Author Response

Response to Reviewer 4 Comments

Point 1: Satellite data are missing. It would be useful to include a table with satellites parameters. Authors stated that images are acquired using different PRF, velocities, burst duration, but values are not mentioned at the beginning of the paper. They are mentioned in Experimental results, but they belong in section 2. 

Response 1: In the revision, I have added a table with satellite parameters in Section 2. In the revised part, I introduces the different PRFs, velocities, burst durations of interferometric images.

Point 2: For specialist it is clear what is SCAN SAR mode, for no specialist some introduction is missing and goals and advantages/disadvantages should be discussed.

Response 2: At the beginning of Section 2, I have added the introduction and advantages/ disadvantages of the ScanSAR mode.

Point 3: Block diagram of SAR data processing is missing, with focusing algorithm.

Response 3: At the beginning of Section 5, I have added the block diagram of SAR data processing. The block diagram introduces the interferometric processing steps. I have not added focusing processing steps in the diagram, because this paper mainly uses focused images to realize the interferometric processing. So focusing is not a work of this paper. The purpose of the focusing algorithm introduction in Section 2 is to give some signal models related to focusing, and then analyze the interferometric performance based on the signal models.

Point 4: In section 3 authors compensated the interferometric coherence by iterative filtering. What is a gain or increase in the coherence? 0.02 for image shown in Fig. 3?.

Response 4: I have revised the iterative filtering experiment. The coherence increase is 0.05 with rectangular filters and 0.02 with combined filters.

Point 5: Rect. windows are usually the worst one to use. Have you considered to use any other filter windows?.

Response 5: Before revision, the used windows are actually combined windows, which is composed of Hanning windows and rectangular windows. In the revision, as a comparison, I analyses the performance when only rectangular windows are used. For a certainty, other filter windows have better interferometric coherence than the rectangular windows.

Point 6: Could you include atmospheric corrections in the model?

Response 6: I have not included atmospheric corrections in the model, because I do not have enough auxiliary data to support these corrections in my paper’s processing. Atmospheric correction is necessary and I will pay more attention to it.

Point 7: Page 9. Longer burst duration, average burst duration, burst central time, all data are missing.

Response 7: I have rewritten the sentences in this part. Quantitatively, I analyses the relationship in Figure 3. For burst duration and burst central time, it is their relative value and not their absolute value that will influence the interferometric coherence. Thus, I analyses their ratio data between burst durations and burst central times.

Point 8: 5. what is a ground truth? How well the interferometric information fits or describe real scene? A comparison with other sensors is missing?

Response 8Because SRTM data have high precision, I uses the data as a ground truth. SRTM is an interferometric SAR system developed by NASA.  In the revision, I also make a comparision to the SRTM data. I have chosen 10 check points in the DEM to compare the difference between mine and SRTM data. I think this comparison can reflect the DEM difference quantitatively.

Thank you very much!

Round 2

Reviewer 2 Report

1, DEM generation from the ScanSAR mode in this paper cannot be accepted by the users. The authors should enhance the value of the proposed method. 
2, Although authors made some improvements in the manuscripts, the derived DEM still has the ripples. These must be removed in a published paper.

Author Response

Response to Reviewer 2 Comments

Point 1: DEM generation from the ScanSAR mode in this paper cannot be accepted by the users. The authors should enhance the value of the proposed method.

Response 1: I have revised the DEM generation method (Page 10). In the revision,  we considered the method to locate the GCPs in the phase image, which is not discussed in original manuscript. Because the GCP informations are their three-dimensional coordinates in the geodetic coordinates system, we need to find the positions of GCPs in the phase image before the geolocation processing. The locating method we used comprises  three calculations for achieving azimuth time, and each calculation is a closed-form solution. So the calculation is efficient. With the discussion of this method, the whole DEM generation method will be more intact.

.

Point 2: Although authors made some improvements in the manuscripts, the derived DEM still has the ripples. These must be removed in a published paper.

Response 2: I have revised the DEM (Figure 8 and Figure 10). These ripples are because of burst unsynchronization. These ripples are located at the areas where different bursts intersect. The burst central time difference in these areas nears the burst cycle time. Thus, the coherence in these areas is 0 and normal interferometric phase stripes cannot be formed. This problem can be overcome by aligning bursts between the master and slave images before splicing the ScanSAR burst images. This aligning method is the best method. But because we cannot obtain the interferometric images before bursts splicing, interpolation method can be used to fill up the invalid areas. So we used the interpolation method to remove most of the ripples.

Thank you very much!

Reviewer 4 Report

Authors answered to all of my questions. 

Author Response

Response to Reviewer 4 Comments

Point 1: Authors answered to all of my questions.  

Response 1: In the revisions of round 2, I have mainly revised my manuscript in two aspects:

I haverevised the DEM generation method (Page 10). In the revision,  we considered the method to locate the GCPs in the phase image, which is not discussed in original manuscript. Because the GCP informations are their three-dimensional coordinates in the geodetic coordinates system, we need to find the positions of GCPs in the phase image before the geolocation processing. The locating method we used comprises  three calculations for achieving azimuth time, and each calculation is a closed-form solution. So the calculation is efficient. With the discussion of this method, the whole DEM generation method will be more intact. I have revised the DEM to remove ripples (Figure 8 and Figure 10). These ripples are because of burst unsynchronization. These ripples are located at the areas where different bursts intersect. The burst central time difference in these areas nears the burst cycle time. Thus, the coherence in these areas is 0 and normal interferometric phase stripes cannot be formed. This problemcan be overcome by aligning bursts between the master and slave images before splicing the ScanSAR burst This aligning method is the best method. But because we cannot obtain the interferometric images before bursts splicing, interpolation method can be used to fill up the invalid areas. So we used the interpolation method to remove most of the ripples.

Thank you very much!